# Factors Affecting Suicidal Thoughts in Breast Cancer Patients

**DOI:** 10.3390/medicina58070863

**Published:** 2022-06-28

**Authors:** Jurgita Kazlauskiene, Alvydas Navickas, Sigita Lesinskiene, Giedre Bulotiene

**Affiliations:** 1Faculty of Health Care, Vilnius University of Applied Sciences, Saltoniskiu Str. 58, LT-08105 Vilnius, Lithuania; j.kazlauskiene@spf.viko.lt; 2Clinic of Psychiatry, Institute of Clinical Medicine, Faculty of Medicine, Vilnius University, M. K. Ciurlionio Str. 21/27, LT-03101 Vilnius, Lithuania; alvydas.navickas@mf.vu.lt (A.N.); sigita.lesinskiene@mf.vu.lt (S.L.); 3Department of Physical Medicine and Rehabilitation, National Cancer Institute, Santariskiu Str. 1, LT-08660 Vilnius, Lithuania

**Keywords:** breast cancer, suicidal ideation, psychosocial factors

## Abstract

*Background and Objectives:* Women diagnosed with breast cancer experience severe trauma. Psychological help for breast cancer patients is not sufficient because of limited professional resources. The goal of this study was to identify groups of breast cancer patients with the greatest suicidal risk, who could be the first target for psychosocial interventions. *Materials and Methods:* The study included 421 women with stage T1–T3/N0–N3/M0 breast cancer. We provided women with a set of questionnaires 1–2 days prior to breast surgery and one year after surgery. One hundred eighty-eight patients completed the questionnaires after one year. We used the Beck Depression Inventory Second Edition (BDI-II) item Suicidal Thoughts or Intentions for the assessment of suicidal risk. The Impact of Event Scale-Revised (IES-R) was used to measure the risk of PTSD and the Vrana–Lauterbach Traumatic Events Questionnaire-Civilian, TEQ-C (TEQ-CV) was used to measure whether patients had experienced other traumatic events in their lifetime. *Results:* The incidence of suicidal ideation one year after surgery increased from 4.3% to 12.8% of patients. Patients who lived in rural areas had a two times greater risk of suicidal thoughts than patients who lived in urban areas. Working patients were 2.5 times more likely to have suicidal thoughts prior to surgery. Severely traumatic events increased the chances of suicidal ideation (OR 7.72; 95% CI 1.63–36.6; *p* = 0.01). The symptoms of PTSD showed a threefold increase in the likelihood of suicidal ideation (OR 2.89; 95% CI 0.98–8.55; *p* = 0.05). *Conclusions:* Living in the countryside, having a history of traumatic experience, having a paid job and having symptoms of post-traumatic stress disorder influence suicidal ideation in breast cancer patients. Particular attention should be drawn to individuals with multiple risk factors.

## 1. Introduction

Oncological diseases are one of the most common causes of mortality in Lithuania and other European Union Member States; they occupy the second place in the structure of mortality. Breast cancer is the most common oncological disease in women worldwide [1,2].

Women diagnosed with breast cancer experience severe trauma [3] that can be manifested in many symptoms [4,5] and affect all aspects of a woman’s life: physical health, work capacity, family life, relationships and psychological well-being [6,7].

Stress, anxiety, hopelessness and depression in breast cancer patients can lead to suicidal thoughts and intentions [8]. A meta-analysis carried out by Franklin and colleagues identified 16 categories of prognostic factors for suicide, including somatic disease, where cancer is one of the subcategories [9]. A study by Bolton and colleagues indicated that oncological disease is the only somatic disease that significantly increases the risk of suicide [10]. Higher risk of suicide is often associated with male gender [11,12], but women are more likely to attempt and to contemplate about suicide [13,14,15]. The prevalence of suicidal ideation among breast cancer patients is significant and varies from 7.8 proc. till 11.4 proc. [13,16,17]. Social-demographic factors influence suicidal thoughts and intentions in breast cancer patients, the most relevant being loneliness [12,18,19], unemployment [12,18], insufficient or unavailable heath care services and age [16,20,21]. Mental health problems, such as depression and posttraumatic stress disorder, are more prevalent among cancer patients than in the general population [22,23,24] and can influence suicidal ideation [25,26]. One of the risk factors that increases the risk of suicide in cancer patients is the time elapsed after diagnosis [27,28]. In Lithuania, patients with early-stage breast cancer complete active anti-cancer treatment an average of nine–twelve months after surgery [29]. Regular contact with the medical profession becomes less frequent, and the patient has to return to his or her social and physical environment. It is at this point that the effects of the illness are most acutely felt [20,30]

It should also be noted that suicide is a major public health problem in Lithuania, where the number of suicides is one of the highest in the world and twice as high as the European Union average [31]. Psychological help for women with breast cancer is not sufficient, and professional resources to help are limited. The extent of the need for psychological help will be revealed having determined the incidence of risk of PTSD, depression and suicidal thoughts among breast cancer patients. The findings of the study will help identify the groups of patients for whom this assistance is most needed. Based on these, the development of the competence of health care professionals could be planned in order to ensure high-quality services that meet the needs of patients. The results obtained can be applied not only to the healthcare of breast cancer patients but also in cases of other oncological diseases. Suicidal ideation associated with trauma caused by the diagnosis and treatment of cancer poses a threat to women’s mental health in the future, and their early identification and assessment is critical. Although psycho-social care for cancer patients is increasingly evolving in many countries around the world, it is still insufficient. Due to insufficient resources for psycho-oncological assistance, it is necessary to prioritize psychosocial assistance for cancer patients.

The goal of this study was to identify the most vulnerable groups of women with breast cancer who could be the first target for psychosocial interventions. The objectives of the study were to determine the frequency of suicidal ideation and its variability over the years in breast cancer patients and to evaluate the factors that influence it.

## 2. Materials and Methods

The study was conducted upon receiving Permit No 158200-07-367-94 of the Lithuanian Bioethics Committee. The study included women with stage T1–T3/N0–N3/M0 breast cancer treated at the Department of Breast Surgery and Oncology of the National Cancer Institute. The number of patients that participated was 421, of which 188 patients completed the questionnaires again one year after the start of the study. The respondents who took part in the study were between the ages of 21 and 80, understood the Lithuanian language and could answer the questions in Lithuanian. Patients who did not receive neo-adjuvant chemotherapy were included in the study. All patients should never have had cancer before, and depression should not have been present for a period of five years.

The Beck Depression Inventory Second Edition (BDI-II) was used to measure suicidal ideation [32]. Green and co-authors concluded that the item Suicidal Thoughts and Wishes included in the BDI-II inventory is appropriate for the assessment of suicidal risk using both 1 and 2 as cut-off values [33].

The Impact of Event Scale-Revised (IES-R) was used to measure the risk of PTSD [34,35]. The assessment of the total IES-R score is the overall indicator of all item scores comprising the methodology. If it is equal to or greater than 34, it indicates the risk of PTSD [34,36,37,38].

The Vrana–Lauterbach Traumatic Events Questionnaire-Civilian, TEQ-C (TEQ-CV) was used to measure whether patients had experienced traumatic events in their lifetime [39].

Patients were asked questions about their social status and were requested to provide sociodemographic data on education, employment, marital status and place of residence. They were also asked when they were diagnosed with the tumor, and whether they were satisfied with how they were notified of the disease. Patients also answered questions about whether they felt the need for psychological help, whether they sought it and where, or whether they had depression for five years.

The study was organized in two phases.

During the first phase, women were provided with a set of questionnaires 1–2 days prior to breast surgery: IES-R, BDI-II and TEQ-CV, and asked questions specifically designed for this study about their social status. Data on patients’ age, diagnosis and treatments used were taken from their medical documentation with the consent of the patients. The questionnaires were filled in by the women themselves; the survey of each woman lasted about 50 min.

The second phase took place one year after surgery. Letters including a questionnaire were sent to the addresses provided by the women. Two of the questionnaires sent were the same as those issued during the first stage: IES-R and BDI-II. The TEQ-CV questionnaire was not used during the second phase, and questions specific to this phase were developed, including the question “Have you experienced any psychologically traumatic events after breast surgery?” Questionnaires were completed independently by women at home and returned by mail to the researcher.

Statistical analysis of the data was performed using the IBM SPSS (Statistical Package for Social Sciences 21 for Windows) software package.

Frequencies and percentage frequencies were calculated for categorical variables. Age was measured on a ratio scale, and variables such as the place of residence, marital status and education were measured on a nominal scale. The mean and standard deviation was calculated at a 95% confidence interval (CI) calculated for continuous variables.

A logistic regression analysis method was used to establish an association between the PTSD risk, depression and suicidal ideation with sociodemographic, clinical and traumatic factors. Standard methods were used to form a model: the Chi square (χ^2^) significance level (*p*) < 0.05; Hosmer–Lemeshow statistics *p* ≥ 0.05; the signs of the coefficients do not seem illogical; at least 50% of cases are classified correctly where Y = 1 and where Y = 0; the coefficient of determination ≥ 0.20. The odds ratio (OR) and its 95% CI was calculated for the final assessment of the conclusions on the relationship between the independent variables and the dependent variable.

Linear regression was used to determine the relationships between PTSD and depression. Standard methods were used to form the model: the correlation of the dependent variable with regressors, ANOVA test *p* < 0.05; the coefficient of determination (R^2^) ≥ 0.2; Cook’s distance values ≤ 1; variance inflation factor ≤ 4. The coefficient B is presented in the event of several regressors and its 95% CI for the final conclusions on the relationship between variables.

The Mann–Whitney and Kruskal–Wallis tests were used to compare non-parametric criteria distributions among independent groups, and Wilcoxon signed-rank and Friedman tests among the dependent ones. The average rank (a non-parametric surrogate for the arithmetic mean indicating which variable tends to be higher) and the sum of the ranks were used to evaluate the variables.

Two-dimensional analyses of the BDI-II and IES-R questionnaire data were performed on a dichotomous scale: *p* is presented according to the McNemar χ^2^ test.

Relationships or differences between criteria were considered statistically significant when the *p* value was lower than the selected significance level α = 0.05 (*p* < 0.05).

## 3. Results

Four hundred twenty-one study patients were included in the study (Table 1). The largest proportion of study patients (50%) were women aged 50 to 64 years. About two thirds of the women participating in the study had less than a bachelor’s degree education. Most of the patients participating in the study were married or had a long-term partner. Unemployed patients accounted for a little less than one third. Three-quarters of the participated patients had lived in a city.

Patients with stage I–III breast cancer (T1–T3/N0–N3/M0) participated in the study according to the criteria for inclusion. Patients diagnosed with stage I accounted for the largest proportion of subjects during both the first (44%) and second (48%) phases of the study. Patients with stage III of breast cancer (19% and 12% for each phase, respectively) accounted for the smallest proportion of subjects. The tumor was detected in half of the patients participating in the study (50.7%) 40 days previously or later, in 29% of patients 40–60 days previously, and in 20% of patients it was three months previously or even earlier.

All patients in the study underwent surgery: 69.6% for mastectomy and 30.4% for breast-conserving surgery (BCS).

Chemotherapy was applied to a little over one third of respondents (37.8%) and radiotherapy to almost half (49.6%) the respondents. Hormone or biologic therapy was applied to 65% of patients.

Suicidal thoughts in breast cancer patients were investigated using the ninth item of the Suicidal Thoughts and Wishes of BDI-II by transforming it into a dichotomous response: 0—I do not have thoughts of suicidal ideation, 1—I sometimes think of suicide. In the first phase of the study (*n* = 421), there were 38, i.e., 9% of patients with suicidal ideation; in the second phase of the study (*n* = 188), there were 24, i.e., 12.8% of patients. The McNemar’s test showed that this item increased significantly among patients who participated in both phases of the study, from 4.3% to 12.8% of patients (McNemar’s test χ^2^ = 8.03, *p* = 0.005).

Factors influencing suicidal ideation in the first phase of the study were investigated (Table 2). A logistic regression model was developed and the effect of sociodemographic factors on suicidal ideation was investigated. Patients who lived in rural areas had a two-times greater risk of suicidal thoughts than patients who lived in urban areas. The likelihood of suicidal thoughts was also increased by having a paid job—working patients were 2.5 times more likely to have suicidal thoughts.

The developed model showed that patients who were satisfied with how they were informed about the disease were less likely to have suicidal thoughts. Patients with a highly traumatic experience were also significantly more likely to have suicidal thoughts. IES-R scores also influenced suicidal ideation. Patients at risk for PTSD were three times more likely to have suicidal ideation than those without risk.

The differences in the influence of different IES-R subscales on the occurrence of suicidal ideation were compared (Table 3). A logistic regression model showed that an increase in the subscale estimates of avoidance and hyper arousal slightly increased the probability of suicidal ideation.

Factors influencing suicidal ideation were investigated in the second phase of the study (Table 4). A developed logistic regression model showed that in the second stage of the study, one year after diagnosis, the probability of suicidal thoughts was reduced by two sociodemographic factors: belonging to the age group over 55 years (OR 0.32; 95% CI 0.12–0.89; *p* = 0.028) and having a paid job in recent years (OR 0.34; 95% CI 0.12–0.94; *p* = 0.037).

We developed a model during the analyses of the influence of emotionally traumatic experience on suicidal ideation, which showed that informing patients about the disease remained important even after a year. Patients who were satisfied with the way they were informed about the cancer were significantly less likely to experience suicidal thoughts (OR 0.18; 95% CI 0.06–0.50; *p* = 0.001). Severely traumatic events increased the chances of suicidal ideation during both the first and second phases of the study by almost eightfold (OR 7.72; 95% CI 1.63–36.6; *p* = 0.01). The risk of PTSD revealed by IES-R also showed a threefold increase in the likelihood of suicidal ideation (OR 2.89; 95% CI 0.98–8.55; *p* = 0.05).

The significance of subscale scores for the occurrence of suicidal ideation was investigated in the second phase of the study (Table 5). The logistic regression model showed that the score of the invasion subscale significantly increased the likelihood of suicidal thoughts.

## 4. Discussion

An emotionally traumatic experience is common in the lives of cancer patients. A higher risk of suicide is usually associated with the male sex, but women are more prone to experience suicidal ideation than men [13]. This could be associated with social roles, sex differences or resilience and neurobiological response to trauma [40]. The fact that Lithuanian women traditionally bear a heavy burden in the family and the asymmetry of family roles is a characteristic that may also be a significant factor [41].

Suicides among cancer patients varies worldwide. Numerous studies analyzing data from cancer registries in European countries [20,27,28,42], the United States [11,30,43], Australia [44] and South Korea [18] demonstrated that the risk of suicide among cancer patients is higher than in the general population. In our investigation, the change in the incidence of suicidal ideation among patients who participated in both phases of the study ranged from 4.3% to 12.9% after one year. The results of the investigation, performed by Anas M Saad et al., showed that the risk of suicide also increases significantly within the first year after a diagnosis of cancer. The highest increase in suicide rates followed diagnoses of pancreatic cancer and lung cancer, whereas a diagnosis of breast cancer did not result in an increased suicide rate [45]. Danish scientist found that suicide risk remained elevated more than 5 years after the diagnosis, if compared to the general population [46].

We analyzed what may cause the presence of suicidal thoughts among Lithuanian breast cancer patients. The place of residence and employment were statistically significant in the sociodemographic factors analyzed in the first phase of the study. Rural life and paid work increased the risk of suicidal thoughts twofold. The patients living in rural areas face more difficult access to health services [47]; moreover, in the general population in Lithuania, suicide is more common in the rural population than in the urban population [48]. Employment remained a statistically significant factor after one year; however, at this stage, paid work had a protective effect against suicidal thoughts. Women who returned to work one year after surgery had a higher material status, more social contacts and better conditions. The living conditions of patients affect immunity and bring strength to fight the disease [24]. The presence of suicidal thoughts was also increased by the younger age of the patient.

Our study demonstrated that the moment when women are provided with information about their cancer diagnosis is very important for them. Receiving a breast cancer diagnosis can be considered as a severe life stressor [49]. More than one third of the patients were not satisfied with the way they were informed about the diagnosis of cancer, and these patients were more prone to suicidal ideation during both the first and second stages of the study. Providing bad news is an issue for both patient and personnel. Regular communication skills training focusing on how to deliver bad news would be helpful for medical staff working with cancer patients [50]. In further investigations, it would be useful to find out useful and handy measures of communication for breast cancer patients.

In both phases of the study, events with severe emotional trauma experienced in the course of life also increased suicidal thoughts. For comparison, it is important to note that the emotionally traumatic experience was also statistically significant for depression, especially if it remained emotionally traumatic. This suggests that intervention that helps to cope with stressful diagnosis should be offered immediately to women after breast cancer diagnoses [49]. The medical personnel must know how to deliver bad news, because it is not only about informing the patient. An important aspect of giving information is not only the need to let the patient know about the cancer diagnosis, but this information must allow patients to participate in the decision-making and treatment process and share responsibility [51].

The risk of PTSD increases the presence of suicidal ideation. The results of the IES-R avoidance and hyper arousal subscales had a significant effect on suicidal ideation in the first phase of the study and intrusions in the second phase, i.e., after one year. This could be related to the fact that if obsessive thoughts about the disease persist for a long time, if they are not expressed and are emotionally unventilated, they can lead to the emergence of suicidal thoughts and thus significantly worsen the well-being of patients.

Limitations of the study. The main limitation of the study was that the majority of the women included in the study were a sample of educated women living in the city. Another limitation was that only women who spoke Lithuanian and could fill in the Lithuanian language questionnaires were included. This makes it difficult to make larger generalizations. To ensure that all influencing factors are considered further research in this area and intercultural comparisons are needed.

## 5. Conclusions

The incidence of suicidal ideation increased one year after breast surgery. Suicidal ideation one year after the surgery was prevalent among patients under 55 years of age. Living in the countryside and paid work caused suicidal thoughts prior to surgery. Severe traumatic experience unrelated to cancer increased the chances of suicidal ideation prior to surgery and one year after surgery. Inadequate delivery of information about the disease can lead to suicidal thoughts.

## 6. Recommendations

We recommend paying particular attention to the psychological problems of breast cancer patients with multiple risk factors.

In order to increase patients’ satisfaction with the delivery of cancer diagnosis, we recommend the strengthening of the communication skills and psychological resilience of physicians and other specialists providing assistance to cancer patients.

## Figures and Tables

**Table 1 medicina-58-00863-t001:** Sociodemographic characteristics of patients (*n* = 421).

Characteristics		Patients Who Participated in the First Phase of the Study *n* = 421	Patients Who Participated in the Second Phase of the Study *n* = 188	Patients Who Did Not Participate in the Second Phase of the Study *n* = 233
Age (years)	Median Range	55 21–80	58 29–80	55 21–80
Education (%)	Less than a bachelor’s degree Bachelor’s degree and higher degree	68.4 31.6	65.9 34.1	70.4 29.6
Marital status (%)	Married or in a long-term partnership Single	62.5 37.5	71.0 29.0	55.1 * 44.9
Employment (%)	Employed Unemployed	28.9 71.1	29.6 70.4	28.4 71.6
Place of residence (%)	City Village	76.6 23.4	77.5 22.5	75.9 24.1

* *p* < 0.05; *n* = number of subjects in the group; *p* = significance level.

**Table 2 medicina-58-00863-t002:** Risk factors for suicidal ideation symptoms before the surgery (*n* = 421).

Variable		B	S.D.	Wald	*p*	OR	95% CI
Residence	0 = Urban 1 = Rural	0.807	0.364	4.915	0.027	2.24	1.1–4.57
Employment	0 = Was unemployed 1 = Was employed	0.937	0.464	4.078	0.043	2.55	1.03–6.34
The constant		−3.246	0.454	51.08	0.001	0.04	
(χ^2^ (2, *n* = 411) = 8.54; *p* = 0.014)							
Was the disease reported to you in the correct manner?	0 = No 1 = Yes	–1.05	0.383	7.52	0.006	0.35	0.17–0.74
The level of impact of the traumatic event	0 = 1–3 points 1 = 4–7 points	2.08	0.750	7.66	0.006	7.98	1.83–34.7
IES-R scores	0 = ≥34 1 = <34	1.24	0.429	8.3	0.004	3.44	1.49–7.99
The constant		–4.21	0.773	29.7	0.001	0.02	
(χ^2^ (3, *n* = 367) = 32.1; *p* < 0.001)							

B—unstandardized regression weight; S.D.—standard deviation; Wald—statistic test; *p*—significance level; OR—odds ratio; CI—confidence interval; χ^2^—chi-squared test; df—degrees of freedom; *n*—cases.

**Table 3 medicina-58-00863-t003:** The influence of IES-R subscales on suicidal ideation before the surgery (*n* = 421).

IES-R Subscale	B	S.D.	Wald	*p*	OR	95% CI
Avoidance	0.07	0.034	4.36	0.037	1.07	1.04–1.15
Intrusion	0.018	0.038	0.228	0.633	1.02	0.95–1.1
Hyperarousal	0.09	0.042	4.57	0.032	1.09	1.02–1.19
The constant	–4.55	0.550	68.6	0.001	0.01	
(χ^2^ (3, *n* = 421) = 35.4; *p* < 0.001)						

IES-R—Impact of Event Scale—Revised; B—unstandardized regression weight; S.D.—standard deviation; Wald—statistic test; *p*—significance level; OR—odds ratio; CI—confidence interval; χ^2^—chi-squared test; df—degrees of freedom; *n*—cases.

**Table 4 medicina-58-00863-t004:** Risk factors for suicidal ideation symptoms after surgery (*n* = 188).

Variable		B	S.D.	Wald	*p*	OR	95 % CI
Age	0 = <55 m. 1 = ≥55 m.	–1.13	0.517	4.8	0.028	0.32	0.12–0.89
Employment over the last year	0 = Was unemployed 1 = Was employed	–1.08	0.518	4.37	0.037	0.34	0.12–0.94
The constant		–0.816	0.453	3.24	0.072	0.44	
(χ^2^ (2, *n* = 184) = 6.47; *p* = 0.039)							
Was the disease reported to you in the correct manner?	0 = No 1 = Yes	–1.72	0.529	10.66	0.001	0.18	0.06–0.51
The level of impact of the traumatic event	0 = 1–3 points 1 = 4–7 points	2.04	0.794	6.63	0.01	7.72	1.63–36.6
IES-R scores *	0 = ≥34 1 = <34	1.06	0.553	3.69	0.05	2.89	1.08–8.55
The constant		–2.99	0.797	14.114	0.001	0.05	
(χ^2^ (3, *n* = 167) = 24.1; *p* < 0.001)							

* Second phase of the study; B—unstandardised regression weight; S.D.—standard deviation; Wald—statistic test; *p*—significance level; OR—odds ratio; CI—confidence interval; χ^2^—chi-squared test; df—degrees of freedom; *n*—cases.

**Table 5 medicina-58-00863-t005:** The influence of IES-R subscales on suicidal ideation after the surgery (*n* = 188).

IES-R Subscale	B	S.D.	Wald	*p*	OR	95% CI	
Avoidance *	–0.021	0.058	0.132	0.717	0.98	0.87–1.1	
Intrusion *	0.203	0.095	4.55	0.033	1.23	1.02–1.48	
Hyperarousal *	–0.17	0.112	2.3	0.129	0.84	0.68–1.05	
The constant	–2.64	0.448	34.7	0.001	0.07		

* after surgery; IES-R—Impact of Event Scale—Revised; B—unstandardized regression weight; S.D.—standard deviation; Wald—statistic test; *p*—significance level; OR—odds ratio; CI—confidence interval; χ^2^—chi-squared test; df—degrees of freedom; *n*—cases.

## Data Availability

The data presented in this study are available on request from the corresponding author.

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
