# Peer review of "Factors Affecting Suicidal Thoughts in Breast Cancer Patients"

_medicina, 2022, doi:10.3390/medicina58070863_

Round 1
Reviewer 1 Report
This is an interesting article, but there are some points that deserve attention.
- Requires English language editor.
- Please clarify the reasons for including breast cancer patients and not patients with other cancer types in the introduction section.
- Please reveal the incidence of suicidal ideation in breast cancer patients in the introduction section.
- Please list the known risk factors for suicidal ideation in breast cancer patients in the introduction section.
- Why were these two time points chosen (1-2 days before breast surgery and one year after surgery)?
- How do you ensure that all influencing factors are considered?
Author Response
Response to Reviewer 1 Comments
Point 1. Requires English language editor.
Response 1. In response to this suggestion, we gave our manuscript to check by a native English-speaking colleague; corrections are highlighted in the text.
Point 2. Please clarify the reasons for including breast cancer patients and not patients with other cancer types in the introduction section.
Response 2. In response to this comment, we have added the following sentence to the Introduction section:“Higher risk of suicide is often associated with male gender (11,12), but women are more likely to attempt and to contemplate about suicide (13–15).” Also we have supplemented the References section accordingly. The entire supplementary reference list is attached (See Additional references).
Point 3. Please reveal the incidence of suicidal ideation in breast cancer patients in the introduction section.
Response 3. In response to this comment, we have added the following sentence to the Introduction section:“The prevalence of suicidal ideation among breast cancer patients is significant and varies from 7.8 proc. till 11.4 proc. (13,16,17).” Also we have supplemented the References section accordingly. The entire supplementary reference list is attached (See Additional references).
Point 4. Please list the known risk factors for suicidal ideation in breast cancer patients in the introduction section.
Response 4. In response to this question, we have added the following sentences to the Introduction section: “Social-demographic factors which influence suicidal thoughts and intentions in breast cancer patients, the most relevant being loneliness (12,18,19), unemployment (12,18), insufficient or unavailable heath care services, age (16,20,21). Mental health problems, such as depression and posttraumatic stress disorder, are more prevalent among cancer patients than in the general population (22–24) and can influence suicidal ideation (25,26).” Also we have supplemented the References section accordingly. The entire supplementary reference list is attached (See Additional references).
Point 5. - Why were these two time points chosen (1-2 days before breast surgery and one year after surgery)?
Response 5.
In response to this question, we have added the following sentences to the Introduction section: “One of the risk factors that increase the risk of suicide in cancer patients is the time elapsed after diagnosis (27,28). In Lithuania, patients with early-stage breast cancer complete active anti-cancer treatment an average of nine - twelve months after surgery (29). Regular contact with the medical profession becomes less frequent, and the patient has to return to his or her social and physical environment. It is at this point that the effects of the illness are most acutely felt (20,30).” Also we have supplemented the References section accordingly. The entire supplementary reference list is attached (See Additional references).
Point 6. How do you ensure that all influencing factors are considered?
Response 5. Thank you very much for this remark. In response to this comment, we have added the following sentence to the Discussion section: “To ensure that all influencing factors are considered further research in this area and intercultural comparisons are needed“.

Reviewer 2 Report
Dear Authors
My overall recommendation- accept in present form.
The article " Factors Affecting Suicidal Thoughts in Breast Cancer Patients" describes a very important problem. In the face of suffering, one should not be indifferent. As the authors rightly said, particular attention should be drawn to individuals with multiple risk factors. This is a very common situation. Living conditions affect immunity and strength to fight the disease. The proposed quotation (one additional) of the following work of publication (Ortenburger D, Rodziewicz-Gruhn J, Wąsik J, Marfina O, Polina N. Selected problems of the relation between pain-immunity and depression, Phys Activ Rev 2017, 5: 74-77. DOI: http://dx.doi.org/10.16926/par.2017.05.10) ) are the result of meritorical reasons. The proposed work for quotation ( in Refferences) is closely related to the subject matter of the paper under review, which includes relevant information. With relation to the fact that the authors of the paper under review may not have been familiar with this paper, I indicated this in the review as a quotation in order to be of benefit to the paper.
Your faithfully
Reviwer
Author Response
Response to Reviewer 2 Comments
Point 1.
The proposed quotation (one additional) of the following work of publication (Ortenburger D, Rodziewicz-Gruhn J, Wąsik J, Marfina O, Polina N. Selected problems of the relation between pain-immunity and depression, Phys Activ Rev 2017, 5: 74-77. DOI: http://dx.doi.org/10.16926/par.2017.05.10) ) are the result of meritorical reasons. The proposed work for quotation (in Refferences) is closely related to the subject matter of the paper under review, which includes relevant information.
Response 1.
Thank you very much for this recommendation. In response to this recommendation, we have added the following sentences to the Discussion section:“Women who returned to work one year after surgery, had a higher material status, more social contacts, and better conditions. The living conditions of patients affect immunity and brings strength to fight the disease (24).”
Also we have added the following reference to the Reference section accordingly:
Ortenburger D, Rodziewicz-Gruhn J, Wąsik J, Marfina O, Polina N. Selected problems of the relation between pain-immunity and depression. Phys Act Rev [Internet]. 2017;5:74–7. Available from:http://dlibra.bg.ajd.czest.pl:8080/dlibra/docmetadata?id=4187&from=publication.
Reviewer 3 Report
The presentation overall is generally very sound.
Some of the sentence structure needs revision. One area that needs more clarity relates to the findings regarding the influence of employment on suicidal ideation in phase 2 of the study. Need to be very clear on the finding and include more discussion on how employment might reduce suicidal ideation in phase 2 of the study.
In addition - more discussion could be given to the critical need for health professions to learn HOW to better engage with patients in supportive ways when delivering sad news.
Author Response
Response to Reviewer 3 Comments
Point 1. Some of the sentence structure needs revision.
Response 1. In response to this suggestion, we gave our manuscript to check by a native English-speaking colleague; corrections are highlighted in the text.
Point 2. One area that needs more clarity relates to the findings regarding the influence of employment on suicidal ideation in phase 2 of the study. Need to be very clear on the finding and include more discussion on how employment might reduce suicidal ideation in phase 2 of the study.
Response 2. In response to this comment, we have added the following sentences to the Discussion section: “Women who returned to work one year after surgery, had a higher material status, more social contacts, and better conditions. The living conditions of patients affect immunity and brings strength to fight the disease (24).”.
Also we have added the following reference to the Reference section accordingly:
Ortenburger D, Rodziewicz-Gruhn J, Wąsik J, Marfina O, Polina N. Selected problems of the relation between pain-immunity and depression. Phys Act Rev [Internet]. 2017;5:74–7. Available from:http://dlibra.bg.ajd.czest.pl:8080/dlibra/docmetadata?id=4187&from=publication.
The entire supplementary reference list is attached (See Additional references).
Point 3. More discussion could be given to the critical need for health professions to learn HOW to better engage with patients in supportive ways when delivering sad news.
Response 3. In response to this reccomendation, we have added the following sentences to the Discussion section: „Providing bad news is an issue for both patient and personnel. Regular communication skills training focusing on how to deliver bad news would be helpful for medical staff working with cancer patients (50). In further investigations, it would be useful to find out useful and handy measures of communication for breast cancer patients.”
Also we have added the following reference to the Reference section accordingly:
Gorniewicz J, Floyd M, Krishnan K, Bishop TW, Tudiver F, Lang F. Breaking bad news to patients with cancer: A randomized control trial of a brief communication skills training module incorporating the stories and preferences of actual patients. Patient Educ Couns [Internet]. 2017 Apr;100(4):655–66. Available from: https://linkinghub.elsevier.com/retrieve/pii/S0738399116305134.
The entire supplementary reference list is attached (See Additional references).

Round 2
Reviewer 1 Report
Good job! The overall quality of the paper has improved, but some improtant references should be cited.
-Fu XL, Qian Y, Jin XH et al. Suicide rates among people with serious mental illness: a systematic review and meta-analysis.
-Du L, Shi HY, Qian Y et al. Association between social support and suicidal ideation in patients with cancer: A systematic review and meta-analysis.